# Study of correlations between serum taurine, thyroid hormones and echocardiographic parameters of systolic function in clinically healthy Golden retrievers fed with commercial diet

**Mara Bagardi**[1,2], **Sara Ghilardi**[1], **Giulietta Minozzi**[1]*, **Eleonora Fusi**[1], **Chiara Locatelli**[1], **Paolo Luigi Ferrari**[3], **Giulia Drago**[3], **Michele Polli**[1], **Elisa Lorenzi**[1], **Francesca Zanchi**[1], **Paola Giuseppina Brambilla**[1]

1 Department of veterinary Medicine and Animal Science, University of Milan, Lodi, Italy, 2 Cardiology Unit, Anicura Clinica Veterinaria Malpensa, Samarate, VA, Italy, 3 Cardiology Unit, Anicura Clinica Veterinaria Orobica, Bergamo, Italy

* giulietta.minozzi@unimi.it

## Abstract

Taurine deficiency predisposes to the development of nutritional dilated cardiomyopathy and is widespread in dogs fed with non-traditional diets. However, Golden retrievers show lower plasma taurine concentration and an impaired systolic function compared to breeds of the same size and morphotype. For these reasons, it can be difficult to classify a subject from a cardiological point of view, with the risk of considering as pathological characteristics that can be completely normal in this breed. This is a cross-sectional multicenter study. The aims were 1) to identify breed-specific range of serum taurine concentration, 2) to describe a correlation between serum taurine concentration and echocardiographic parameters of systolic function in clinically healthy Golden retrievers fed with traditional diet, 3) to identify a correlation between thyroid hormones, serum taurine concentration and echocardiographic indices. Sixty clinically healthy Golden retrievers (33% males, 67% females) were included. Fifty-three dogs were fed with traditional diets and their range of serum taurine concentration was 398.2 (31.8–430) nmol/ml. Serum taurine concentration was found to be negatively correlated to systolic internal diameter of the left ventricle and systolic and diastolic left ventricular indices and volumes obtained with different methods, whereas was positively correlated to the left ventricle ejection and shortening fractions but difference was not statistically significative. A weak but significant correlation between serum taurine and T4 was demonstrated. Serum taurine median values in dogs with normal systolic function were higher than in dogs with impaired systolic function. A cut-off of serum taurine concentration of 140.6 nmol/ml had a moderate sensitivity and specificity in the identification of an impaired left ventricular systolic function (AUC 0.6, Se 78%, Sp 44%). This study showed that the median serum taurine concentration was significantly lower in dogs with impaired systolic function. Therefore, echocardiographic monitoring is recommended in all dogs with serum taurine concentration lower than 140.6 nmol/ml.

**Data Availability Statement:** All relevant data are within the paper and its supporting information files.

**Funding:** The project has been approved by the Anicura Scientific Council in 2022 and the Authors received funding for this project.

**Competing interests:** The authors have declared that no competing interests exist.

## Introduction

Canine dilated cardiomyopathy (DCM) is the second most common acquired heart disease in dogs [1]. Multiple aetiologies have been identified [1]. While DCM of genetic origins have been described in some breeds based upon discovered mutations or observed heritability and pattern of inheritance, determining the aetiology of DCM when observed outside of these breeds is challenging [1, 2]. Nutritionally mediated DCM (NDCM) has been identified in a lot of species, including dog, and is related to taurine deficiency [3–5]. Recent peer reviewed research on DCM in breeds that were not previously known to have a genetic predisposition for the disease has raised concern about the relationship between diets with certain characteristics and the development of NDCM [6–8]. The Food and Drug Administration (FDA) issued a warning and subsequently released data that identified dietary characteristics which were over-represented in some pet food [9]. These data are supported by similar findings from researchers at many institutions and suggest that diets which are grain-free or contain legume or potato ingredients should be study to further elucidate their possible role in the causation of DCM [7–10]. When evaluated, the FDA data also identify an inverse relationship/correlation between the size of a company in terms of worldwide sales and the number of reported cases of DCM (smaller companies have the highest reported cases) [9]. The Golden retriever (GR) is the breed that is most frequently reported to be affected by NDCM in the FDA report [9]. If compared to other breeds eating similar diets, the role of taurine deficiency in this breed appears of great interest. The over-representation of GRs is interesting as there is no literature to support any familial relationship or genetic aetiology for classic DCM in this breed. In 2005 an autosomal recessive and/or polygenic transmission has been supposed, but this theory has not been confirmed in subsequent studies [11]. In 1999, a survey conducted on 1400 GRs belonging to the same breed club showed that the incidence of cardiomyopathy is less than 0.7% [12]. Additionally, a large study on heart diseases in insured dogs identified GRs as a low-risk breed for all cardiac claims and reported that they have a lower cardiac mortality rate than the pooled study population [13]. Thus, the investigators sought to examine the relationship between diet and nutritional elements with NDCM without or with taurine deficiency. Several studies have reported that GRs may be more susceptible to taurine deficiency, supporting the hypothesis that the GRs are particularly sensitive to dietary taurine deficiency, and probably, this breed has a greater requirement of this amino acid than others [14, 15]. These findings are supported by the assessment of lower blood taurine reference ranges in the GRs compared to other breeds: the taurine ranges on whole blood and plasma are respectively 214–377 nmol/ml and 63–194 nmol/ml in healthy GRs against the ranges of 261–271 nmol/ml and 75–79 nmol/ml in healthy dogs of different breeds [15]. Golden retrievers fed with diets that directly or indirectly affect taurine blood concentration (e.g., grain-free diets, low protein and amino acid diets, high fibre, or legume content) often appear to present hypotaurinemia and more easily develop forms of DCM [14, 15]. The authors have previously reported a serum taurine plasma concentration in 10 GRs that was lower than the value reported in another large breed (German pointer, not reported as predisposed to taurine deficiency) in a pilot study carried out in 2021 [16]. Moreover, it has been demonstrated that this breed normally tends to have higher ventricular volumes, lower sphericity index, and lower ejection fraction when compared to the general canine population [16]. These breed-specific echocardiographic features should be taken into consideration for an accurate echocardiographic interpretation and screening every time the cardiologist must evaluate one of these subjects. The clinician must also keep in mind that this breed is predisposed to an increased risk of developing hypothyroidism, for which an association with an impaired left ventricular systolic function has not yet been completely demonstrated [17–20]. This introduces new doubts on myocardial disease

with DCM echocardiographic phenotype in GRs. For all these reasons, the presence of an association between taurine deficiency, thyroid hormones, diet and dietary ingredients and the left ventricular systolic function in the myocardial disease process in this breed must be investigated to avoid the risk of interpreting as abnormal something that could be normal for the breed. In addition, to gain some insight into the possible cause of DCM in a not genetically predisposed breed, a detailed study on the mode of inheritance, also including the inbreeding evaluation, need to be considered.

The hypothesises of this study are that: 1) Clinically healthy GRs with decreased systolic function have lower serum taurine concentration, 2) GRs with lower serum taurine concentration have higher inbreeding and kinship scores.

Therefore, the aims of this study are: 1) To identify the relationship between serum taurine concentration, T4 and TSH assessment and echocardiographic indices of systolic function in clinically healthy GRs, 2) To establish a cut-off of serum taurine concentration that can discriminate between healthy GRs and GRs with systolic dysfunction, 3) To perform an inbreeding and kinship check through the pedigree evaluation.

## Materials and methods

Golden retrievers were recruited for this cross-sectional multicenter study from the populations of the cardiovascular breed screening examinations at the Cardiac Unit of the University veterinary teaching hospital in Lodi—University of Milan—Italy and at Anicura Clinica Veterinaria Orobica in Bergamo—Italy between July 2022 and June 2023. The possibility to take part on this research has been presented to the owners of the dogs recognized as possible participants. The study was approved by the Anicura Committee and all study participants provided written informed consent.

Inclusion criteria were an unchanged diet history for at least 3 months prior to enrolment, a complete echocardiographic examination without any pharmacological restraint, and a complete diet history. All included GRs were subjected to a complete physical examination, systemic blood pressure measurement via Doppler method, electrocardiogram, and echocardiography. Dogs were considered healthy basing on the absence of prior clinical conditions/abnormalities documented by the owners and unremarkable physical examination, blood analysis, and cardiovascular assessment. Blood samples were processed (complete blood count, biochemical profile, complete thyroid profile, urinary analysis, and troponin I concentration) and considered normal according to the laboratory reference ranges. Dogs with congenital heart diseases were excluded from the study, as well as dogs aged <18 months because of the possible influence of young age on the serum taurine concentration.

### Diet history

Traditional diets (TD) were required to meet the following criteria: kibble diets which are grain-inclusive and not include legumes or potatoes in the top 5 ingredients listed. Non-traditional diets (NTD) had to contain kibble or raw food diets that are grain-free or include legumes or potatoes [9]. Body condition score was assessed by the veterinary nutritionist (EF) and recorded using a validated 9-point scale [21]. Dogs with diet histories not meeting the defined categories were excluded (i.e., dogs eating a mix from the TD and NTD group, etc.). Dogs with incomplete diet histories that could not be elucidated through a contact with the owner were excluded. Dogs that did not have a consistent diet history for >3 months were excluded. Dogs receiving dietary supplements containing taurine, methionine or L-carnitine have been excluded.

## Blood analysis

Venous blood samples were obtained for complete blood count, complete biochemical analysis, serum thyroid profile ($T_4$ and TSH), serum taurine concentration measurements, and plasma troponin (cTnI). Fasting was required before blood sampling, even if fasting status does not impact taurine level in dogs; however, it impacts biochemical analysis [22] and cTnI concentration [23]. Complete blood count (CBC), associated with the evaluation of the blood smear and the biochemical analysis, was performed at the internal laboratory of the University Veterinary Teaching Hospital of the University of Milan (Lodi) with the automated hematology analyzer Sysmex XT-2000iV (Sysmex, Kobe, Japan). Biochemical analyses were carried out with the automated spectrophotometer BT3500 (Biotecnica Instruments, Rome, Italy). The dosage of serum taurine was performed at the San Marco veterinary laboratory (Padova—Italy) by liquid chromatography with a non-contact mass spectrophotometer (LC-MS/MS, liquid chromatography-mass Spectrometry), a method developed and validated by the certified laboratory itself. Evaluation of the thyroid profile was conducted at the Idexx laboratory through chemiluminescent immunoassay and enzyme immunoassay for TSH and T4 respectively. Measurement of cTnI was carried out using the Immulite (EuroDPC, Gwynedd, Wales) according to the manufacturer's recommended methods. The manufacturer's quoted imprecision was 8.4–6.1% (0.8–86 μg/L). The detection limit of the assay was 0.1 μg/L, the 99th centile of a reference population was 0.2 μg/L, and the upper measurement upper limit was 180 μg/L. The total imprecision of the assay was evaluated using NCCLS protocol EP-5A and compared with the manufacturers claims of 8.4–6.1% across the range 0.8–34 μg/L [24].

## Urine analysis

All urine samples were collected through cystocentesis and were immediately refrigerated. Within 8 h, standard urinalysis was performed by dipstick chemistry test and refractometer (for USG evaluation); all samples were then immediately centrifuged at 1250 rpm for 5 min and the supernatant was stored at − 20 ˚C. The supernatant underwent urinary protein (UP) and urinary creatinine (UC) evaluation by Pyrogallol Red Method and UP/UC was calculated (values < 0.5 were considered normal [25]).

## Echocardiography

All dogs received an echocardiogram by an experienced echocardiographer or a cardiologist resident in training, both blinded to diet history and assigned diet group, using three ultrasonographic units equipped with two different multifrequency phased array probes (Esaote MyLab[TM] 30 Gold, Esaote MyLab[TM] Omega, Mindray M9). Images were also stored for off-line analysis. All echocardiographic scans were carried out on conscious dogs in right and left lateral recumbency, in accordance with previous published standards [26, 27]. All measurements were taken from three consecutive cardiac cycles, and the mean was recorded. Echocardiographic measures recorded for the study included LA/Ao measured in 2D-mode using the Hansson's method [28] from the right parasternal short axis view, measurement of left ventricular internal diameter in diastole (LVIDd), left ventricular internal diameter in systole (LVIDs) and calculated percent fractional shortening (SF) and the ejection fraction (EF). Left ventricular measurements were performed from the right parasternal short-axis view with M-mode of the left ventricle at the level of the papillary muscles with the leading edge to leading edge method. Fractional shortening was calculated according to the equation: (LVIDd-LVIDs)/LVIDd x 100. Ejection fraction was calculated from M-mode and B-mode measurements using the Teichholz method to determine chamber volumes as previously published [29]. End diastolic volume index (EDVI) and end-systolic volume index (ESVI) were also

calculated by dividing the end-systolic or end-diastolic volumes in millilitres by body surface area in squared meters as previously described [30]. Body surface area was calculated by the formula 0.101 x body weight $(kg)^{2/3}$. Normalized left ventricular end-diastolic and end-systolic internal diameters (LVIDNd and LVIDNs) were obtained using the allometric equation, as previously described [31]. LVIDd was considered increased when >51 mm [32], and LVIDs was considered increased when >35 mm [32]. The fractional shortening was recorded as low when <25% [31, 32]. E-point septal separation (EPSS) was obtained from M-mode measurements in the right parasternal short axis view at the mitral valve level [33]. The mitral and tricuspid annular plane systolic excursion (MAPSE and TAPSE) were measured according to the literature form the left apical four chambers view with M-mode [34, 35]. End-systolic and end-diastolic left ventricle volumes (LVVs e LVVd) were obtained from B-mode measurements by the Simpson method and by the area-length method in the right parasternal long axis view. Left ventricle sphericity index (SI) was calculated from B-mode measurements in the left parasternal apical 4-chambers view [36]. Fractional area change (FAC%) of the left ventricle was calculated from B-mode measurements in the right parasternal short axis view at the papillary muscles level. Left ventricular diastolic diameter to aorta ratio (LVIDd/Ao) was obtained from B-mode measurements in the right parasternal long axis view. Transmitral flow (E peak velocity, A peak velocity, E peak velocity-to-A peak velocity ratio) was measured using pulsed-wave Doppler from the left four chamber apical view. Aortic and pulmonary flows were also evaluated from the subxiphoid view and from the right parasternal short axis view at the base of the heart, respectively. The chosen cut-off values were based on previously published data and on breed-specific reference intervals to maintain consistency with prior publications [13, 31, 32, 37].

**Systolic function evaluation.** The subjects were classified as dogs affected by impaired systolic function according to the following criteria: 1) LVVd >101.6 $cm^2$ and LVIDNd >1.7 or LVVd >101.6 $cm^2$ and sphericity index <1.65 or LVIDNd >1.7 and sphericity index <1.65; 2) at least two of these parameters: EF <40%, LVVs >45.6 $cm^2$, SF <20%, FAC < 35%, EPSS > 0.7 cm. Dogs with normal systolic function presented: LVVd and LVVs included in the normal reference ranges for weight (41.1–106.1 $cm^2$ and 18.4–45.6 $cm^2$), LVIDNd ≤1.7, sphericity index ≥1.65, EF ≥40%, FS ≥35%, FAC ≥35% and EPSS ≤0.7 cm. All dogs with echocardiographic characteristics that did not meet these criteria were not included in the analysis of the assessment of systolic function.

## Pedigree analysis

Pedigree information were used to estimated inbreeding coefficients on the entire cohort. Data handling and calculations were performed in the R statistical environment [38], version 4.1.3 with the "OptiSel" package [39]. The "prePed" function of the "OptiSel" package was used to prepare the pedigree file. The function "summary.Pedig" was used to calculate the pedigree inbreeding coefficient of all individuals that had pedigree information, the number of fully traced generations and the number of maximum generations traced [40, 41]. The analysis was conducted both on the entire cohort and subsequently means of inbreeding were given only on the animals with impaired systolic function.

## Statistical analysis

Statistical analysis was performed using SPSS 28 (IBM, SPSS, USA). Data were tested for normality using the Shapiro–Wilk test. Normally distributed data were presented as mean ± standard deviation (SD) and compared by the two-sided Student's t-test and non-normally distributed data were presented as median and interquartile range (IQR) and compared

by the median test. Correlation was tested by the Pearson rho ($\rho$) correlation coefficient, with the following interpretation: $\leq 0.3$ weak correlation, $> 0.3$ and $\leq 0.7$ moderate correlation, $> 0.7$ strong correlation. Requiring 80% power, weak correlations less than 0.35 were considered for descriptive purposes only.

Multiple linear regression was performed, and the backward method was used. Only an R square value greater than 0.1 (P<0.05) was considered suitable.

Receiver operating characteristic (ROC) analysis was performed. The area under the ROC curve (AUC) was used to assess the diagnostic accuracy and to quantify the predictive value of serum taurine concentration for systolic dysfunction, as suggested by Šimundić in 2009 [42]; the degree of diagnostic accuracy was interpreted as follows: non informative test (AUC = 0.5); inaccurate test ($0.5 <$ AUC $\leq 0.7$), accurate test ($0.7 <$ AUC $<1$), perfect test (AUC = 1) [43]. A cut-off value was found by maximizing the Youden index. Statistical significance was accepted at p-value $\leq 0.05$.

## Results

The study included 60 Golden Retrievers, 40 females (66.7%), 3 spayed (5%), and 20 males (33.3%), 2 neutered (3.3%). The mean age was 4.2±2.8 years and the mean weight was 30.2±4.7 Kg. The mean heart rate, systolic, diastolic, and mean blood pressure were 101.3±14.4 bpm, 136.0±17.3 mmHg, 79.4±19.9 mmHg and 101.1±15.9 mmHg, respectively. All the pressure data were considered normal according to the guidelines for dogs and cats. The mean BCS was 5.2±0.8 points. Fifty-three (88.3%) dogs were fed with TD, 5 (8.3%) with GF diet, and 2 (3.4%) with NTD. Furthermore, forty-five of these diets were gluten free. The echocardiographic data of the included population are reported in Table 1.

Dogs with normal systolic function were 34 (56.7%), while 14 (23.3%) presented impaired systolic function (8–13.3%–according to the first criterion and 7–11.7%–to the second one). The other 12 (20%) dogs presented a systolic function not strictly classifiable according to the criteria reported in the materials and methods section and therefore were not considered for the analysis.

The urinalysis was normal for all included dogs, including the UP/UC ratio. The median serum taurine concentration in the overall population was 121.1 (78.2–171.9) nmol/ml. There were no statistical differences in the serum taurine concentrations of dogs with different sex and weight.

The serum taurine concentration levels of the 48 subjects meeting the inclusion criteria for the analysis of the systolic function were related to the echocardiographic parameters. A negative weak to moderate correlation with LVIDs ($\rho$ = -0.345; P = 0.009), EDVI ($\rho$ = -0.273; P = 0.042), ESVI ($\rho$ = -0.361; P = 0.006), EPSS ($\rho$ = -0.277; P = 0.040), LVVd, and LVVs ($\rho$ = -0.406; P = 0.003 and $\rho$ = -0.385; P = 0.006, respectively) was described. Whereas, between taurine, EF ($\rho$ = 0.360; P = 0.006) and SF ($\rho$ = 0.352; P = 0.008) a positive moderate correlation was found. Fig 1 shows the scatter plots of echocardiographic variables whose correlation absolute value with serum taurine was stronger than 0.35.

In addition, the study of normality allowed to identify dogs with echocardiographic parameters of impaired systolic function as having a lower median serum taurine concentration compared to normal dogs (100.88 versus 131.36 nmol/ml), however the result was not statistically proved (P>0.05). The diagnostic accuracy of optimal serum taurine concentration cut-off (140.6 nmol/ml) for prediction of the reduction of systolic function was reported, as well as sensitivity (78%) and specificity (44%) (AUC = 0.60) (Fig 2).

By reducing the serum taurine cut-off (124.85 nmol/ml), the sensitivity decreased (71%) and the specificity increased (59%).

Table 1. Echocardiographic measurements of the included population.

| | Mean | SD | Median | IQR$_{25}$ | IQR$_{75}$ | Minimum | Maximum |
|---|---|---|---|---|---|---|---|
| LA/Ao | 1.32 | 0.16 | | | | 0.99 | 1.86 |
| LVIDd (mm) | 42.4 | 4.1 | | | | 30.3 | 52.4 |
| LVIDs (mm) | 29.8 | 4.4 | | | | 20.0 | 39.7 |
| LVIDNd | 1.48 | 0.15 | | | | 1.14 | 1.92 |
| LVIDNs | 1.03 | 0.16 | | | | 0.71 | 1.38 |
| LVIDNd/Ao | 2.50 | 0.35 | | | | 1.74 | 3.47 |
| EDVI (mL/m$^2$)* | | | 82.6 | 73.5 | 90.5 | 42.8 | 149.9 |
| ESVI (mL/m$^2$) | 36.9 | 13.3 | | | | 14.0 | 72.2 |
| EF (%) | 56.4 | 10.0 | | | | 30.0 | 82.0 |
| SF (%) | 29.6 | 6.8 | | | | 14.0 | 51.0 |
| EPSS (cm)* | | | 0.47 | 0.40 | 0.58 | 0.29 | 0.92 |
| MAPSE (mm) | 13.3 | 2.6 | | | | 8.8 | 19.6 |
| TAPSE (mm) | 14.87 | 2.6 | | | | 7.8 | 20.2 |
| SI* | | | 1.5 | 1.4 | 1.7 | 1.1 | 2.2 |
| FAC (%) | 42.8 | 7.0 | | | | 29.5 | 64.4 |
| LVVd-SMOD (cm$^2$) | 58.7 | 14.3 | | | | 27.4 | 95.7 |
| LVVs-SMOD (cm$^2$)* | | | 23.9 | 16.6 | 32.9 | 11.7 | 47.0 |
| E/A | 1.33 | 0.31 | | | | 0.69 | 2.05 |
| Aortic peak velocity (m/s) | 1.25 | 0.36 | | | | 0.58 | 2.21 |
| Aortic peak gradient (mmHg) | 6.7 | 3.8 | | | | 1.3 | 19.5 |
| Pulmonary peak velocity (m/s) | 0.78 | 0.18 | | | | 0.44 | 1.30 |
| Pulmonary peak gradient (mmHg) | 2.58 | 1.20 | | | | 0.80 | 6.78 |

* Non normally distributed variable. LA/Ao, ratio between left atrium diameter and aortic diameter; LVIDd and LVIDs, left ventricular internal diameter in diastole and systole obtained with M-mode evaluation; LVIDNd and LVIDNs, normalized left ventricular internal diameter in diastole and systole (allometric method); LVIDNd/Ao, ration between normalized left ventricular internal diameter in diastole and aortic root; EDVI and ESVI, end-diastolic and end-systolic volume indexes obtained with Cornell method; EF, ejection fraction of the left ventricle obtained with Simpson method of discs; SF, shortening fraction of the left ventricle obtained with Simpson method of discs; EPSS, E point to septal separation obtained with M-mode evaluation; MAPSE, mitralic annular plane systolic excursion obtained with M-mode evaluation; TAPSE, tricuspid annular plane systolic excursion obtained with M-mode evaluation; SI, sphericity index obtained with B-mode evaluation; FAC, fractional area change of the left ventricle obtained with B-mode evaluation; LVVd-SMOD and LVVs-SMOD, left ventricle volumes in diastole and systole obtained with Simpson method of discs (B-mode); E/A, transmitral E and A waves velocity ratio.

Regarding the thyroid profile, none of the included subjects presented hypothyroidism. The median TSH was 0.09 (0.07–0.13) ng/ml and the mean T4 was 1.63±0.63 μg/ml. Although a weak correlation between serum taurine concentration and T4 was found (ρ = 0.3; P = 0.05), no correlation with TSH was proved. The T4 was not statistically different in subjects with impaired systolic function.

Results of the complete blood count (CBC) and biochemical analysis are shown in the S1 Table. All the results were within reference ranges. The statistical analysis was performed to evaluate the correlation between CBC, biochemical analysis, and serum taurine concentration, but no statistical significance was identified (P>0.05). The same was observed for the correlation between blood results and echocardiographic parameters suggestive of impaired systolic function (P>0.05).

Results of the pedigree analysis shown in the S2 Table have been estimated on 50 of the 60 animals in the study. Ten dogs were not included as no pedigree information was available. The mean inbreeding coefficient of the 50 dogs was 0.031±0.058, while the highest value was 0.298 and the lowest value was 0. Subsequently estimates were calculated only on the 14 dogs

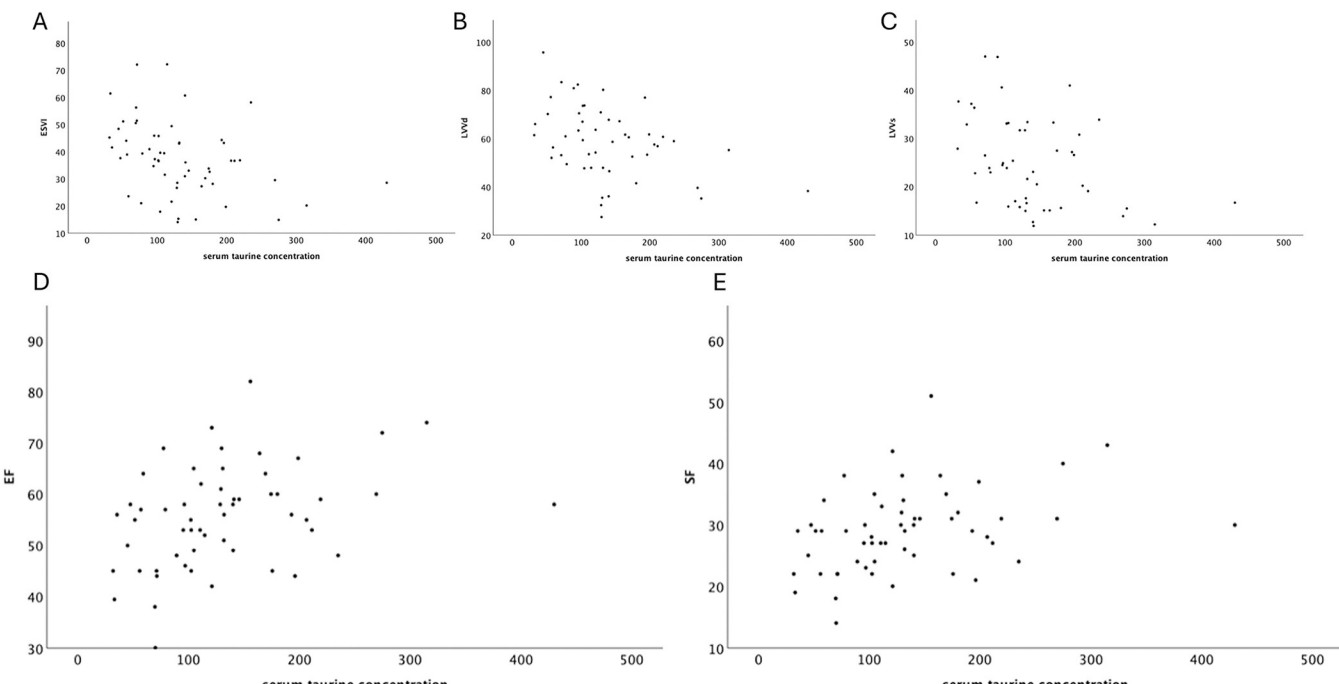

**Fig 1.** A-C Scatter plots showing the negative moderate correlation between serum taurine concentration and ESVI, LVVd, and LVVs, respectively. D-E Scatter plots showing the positive moderate correlation between serum taurine concentration and EF and SF.

with altered systolic function. Inbreeding coefficients were available only for 13 of them due to pedigree availability. The mean inbreeding of this group was estimated to be 0.043 ± 0.076, with highest value 0.289 and lowest value 0.001. However, inbreeding coefficient in the two groups did not differ significantly (P = 0.278).

## Discussion

Among the canine DCM etiologies, the literature reports the blood taurine deficiency [1–5]. The Golden retriever is the most represented breed affected by this peculiar form of DCM, despite this breed is not predisposed to the development of the hereditary form of DCM [9, 11–13]. Furthermore, the literature reports that the Golden Retriever has lower basal taurine levels compared to the other breeds and for this reason it could be more sensitive to an additional deficiency state [13–15]. Some studies reported the reference ranges for plasma, and whole blood taurine in this breed [13–15]. The present study confirmed the lower taurine levels compared to serum concentration reported for other breeds, showing a median serum taurine concentration of 121 (78.2–171.9) nmol/ml. Since the concentration of taurine varies greatly depending on the substrate analyzed, further studies on the evaluation of a breed-specific reference range are needed, because the literature focused on the evaluation of plasma and whole blood ranges, limiting the data regarding the serum concentration. A cause of hypotaurinemia in Golden Retrievers has not yet been clarified. A hereditary-genetic basis has been supposed [10]. However, it is also suspected that the low taurine level is not a primary condition but could be due to other typical alterations of this breed, such as intestinal dismicrobism. An altered intestinal microbiome can cause an absorption modification of this amino acid, thus hesitating in a deficiency state. Further studies are required to confirm this supposition in this breed.

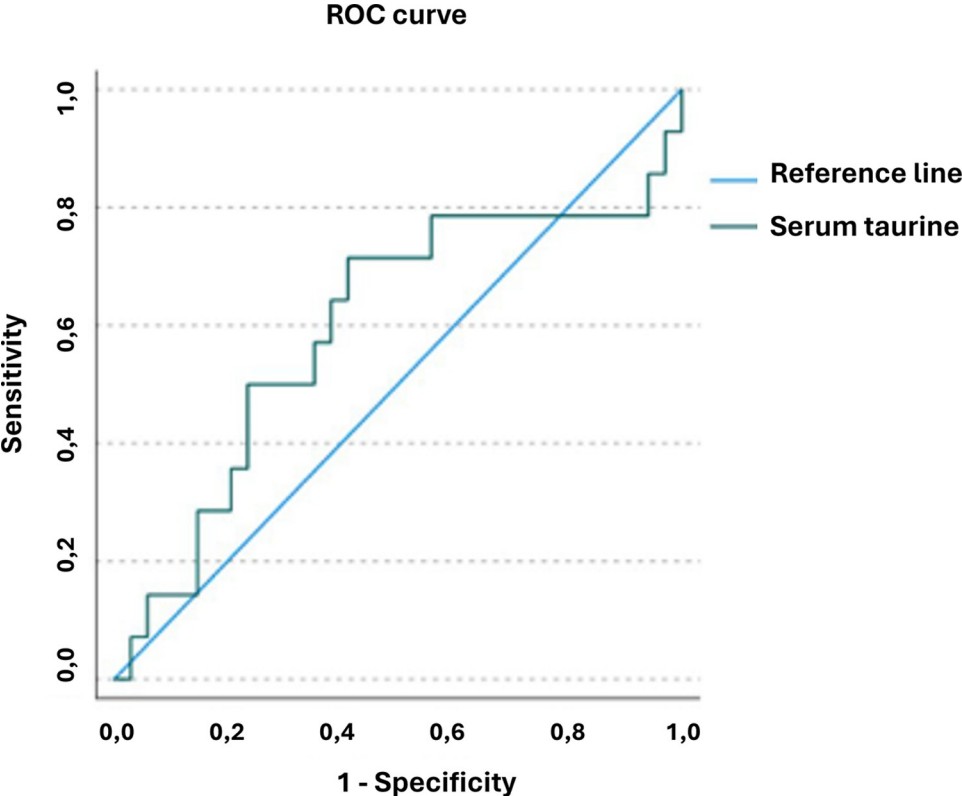

**Fig 2. Taurine ROC curve for the identification of impaired systolic function.**

However, it is also suspected that the taurine metabolism could be altered due to intestinal dysbiosis. It is well known that taurine is processed by intestinal bacteria via bile salt hydroxylase enzymes and dehydroxylation to yield secondary bile acids. Dogs affected by IBD showed higher faecal concentrations of primary bile acids that have been correlated with a lower expression of apical sodium-dependent bile acid transporter proteins in the ileum [44]. Moreover, in their study on human pro-carcinogenic gut microbiota Yang and co-authors (2023) evidenced a reduction of serum taurine associated with the increase in Desulfovibrionaceae and the decrease in Lactobacillus in the intestinal community [45].

This study demonstrated that sex and body weight were not related to taurine concentration. To the author knowledge, no studies reported the correlation between serum taurine concentration and echocardiographic data, in particular those measurements describing the left ventricular systolic function. This study focused on the evaluation of these parameters and found a significant correlation between serum taurine concentration and LVIDs, ESVI and EDVI, LVVs and LVVd obtained with Simpson method of discs, EPSS, EF and SF. Only EF and FS showed a positive correlation with taurine. These correlations suggest that low serum taurine concentration is associated to impaired systolic function in clinically healthy Golden Retrievers. The included subjects were classified into two subpopulations basing on systolic function indices (normal vs subjects with impaired systolic function). These two populations had a different median and distribution of serum taurine concentration. The group with impaired systolic function showed a lower median taurine concentration compared to the other group (100.88 nmol/ml vs 131.36 nmol/ml). It is interesting to underline how the taurine distribution in dogs with decreased systolic function showed a decreasing trend, with data

more focused on the lower limits of the group range, whereas in the group of normal dogs the taurine concentration focused more on the mean value. Therefore, the obtained results agree with studies reporting a relationship between taurine deficiency and the development of DCM [3–8]. Based on this, it is possible to emphasize the importance of a correct and thorough dietary history in subjects suspected of DCM to set a correction of the diet and an oral integration of this amino acid. This study has also defined, through the evaluation of a ROC curve, a cut-off of serum taurine concentration that can be clinically useful for the identification of a systolic function reduction (a cut-off of 140.6 nmol/ml showed a sensitivity of 78% and a specificity of 44%). Although in the literature it has been shown that a state of hypotaurinemia is related with the intake of specific diets, such as NTD, grain free diets or diets containing legumes and potatoes [9], this study was not able to demonstrate it, since the population was too small and only few subjects were fed constantly with one of the diets listed above. This study also focused on the evaluation of the thyroid profile, as a reduction of thyroid hormones is reported in the literature as an additional possible etiology of DCM [1, 16].

In addition, the Golden Retriever is described as one of the breeds predisposed to the development of hypothyroidism [16]. The subjects included in the study were not hypothyroid. The results showed a weak correlation between serum taurine concentration and T4, but no significant correlation with TSH. Therefore, this study did not demonstrate a real correlation between hypotaurinemia and changes in the thyroid profile. These results are part of a heterogeneous literature describing an effective correlation between hypothyroidism and DCM development. Although this link is proven in humans, in veterinary medicine the results are still confusing. Some of them reported an improvement in the systolic function after thyroxine intake, while others showed that, although hypothyroid subjects show cardiac changes, a specific correlation between systolic dysfunction and hypothyroidism is not demonstrable, clarifying that the described cardiac modifications are not able to determine an overt DCM [17–20].

Furthermore, as pedigree information of the dogs was available, estimates of inbreeding coefficients have been produced. There was no significant difference in the mean of the inbreeding coefficient between the dogs with healthy or impaired systolic function. However, the two groups were not homogenous and further analysis should be conducted on a more balanced cohort.

This study had some limitations. The first is the lack of the evaluation of intra and inter-operator variability of echocardiographic measurements. The different level of experience and the use of different ultrasound machines may have introduced some analytical errors, not estimated. However, the Osservatorio Veterinario Italiano Cardiopatie (OVIC) accredited veterinarians as well as operators working at the University Veterinary teaching hospital, have an intra-operator and inter-operator variability ≤8 and ≤12% respectively, depending on the echocardiographic measurement, as demonstrated by previous variability tests. Another limit is the inclusion of a population with a small number of subjects fed with grain free and with NTD. This did not permit to assess if the serum taurine concentration in these subjects was lower compared to dogs fed with TD, thus confirming an already suggested correlation between taurine level and type of diet. Moreover, the sample was very homogeneous in terms of age, with predominantly young-adult subjects. This may have affected the levels of thyroid hormones, that were within the normal range for the included population, as endocrine diseases are mostly related to senility. This may have affected the lack of a correlation between serum taurine levels and thyroid hormones. Finally, an important limitation was the lack of the follow-up of the included dogs. For this reason, the possible development of systolic dysfunction and DCM are not known.

## Conclusion

The results of this study showed a correlation between serum taurine concentration and echocardiographic parameters of left ventricular systolic function. Lower serum taurine concentrations are related to larger left ventricle diameters and volumes (both systolic and diastolic), as well as greater EPSS, and lower ejection and shortening fractions. This demonstrated that low serum taurine concentration was associated with impaired left ventricular systolic function. The results of this study suggest that the evaluation of the serum taurine concentration should be performed in normal (or healthy) GRs with impaired systolic function at the echocardiographic examination. For this reason, the results of this study suggest that an echocardiographic examination when serum taurine concentration is lower than the proposed cut-off of 140.6 nmol/ml should be performed. Finally, the study showed a weak correlation between echocardiographic indices of systolic function and T4, but not with TSH. No difference in thyroid profile was observed in normal subjects compared to those with impaired systolic function. Basing on the obtained results, it would be interesting to evaluate the correlation between serum taurine concentration, echocardiographic indices of left ventricular systolic function and thyroid profile in a larger sample of dogs, with greater variability in terms of age, including older subjects, more likely affected by an alteration of the thyroid profile. A study with a larger number of subjects fed with NTD is also desirable to further investigate the correlation between hypotaurinemia and the type of diet. In addition, research is already in progress to trace a possible link between serum taurine deficiency associated with intestinal dysbiosis: an alteration of the intestinal microbiome may be the cause of a lack of absorption of taurine in GRs, generating a deficiency state. Since there are no previous studies on a possible predisposition of the Golden Retriever to intestinal dysbiosis, it remains to clarify whether the hypotaurinemia is a primary condition and is related to the breed or if it is secondary to other pathological condition of the GR. Since DCM has proven strong hereditary bases in different large breed dogs, its prevalence is also linked to the degree of inbreeding within a population. In this study, for 45 subjects a pedigree analysis was available, and a very limited inbreeding coefficient was found, evidence that the obtained results were not affected by consanguinity.

## Supporting information

**S1 Table. Results of the complete blood count and biochemical analysis of the dogs included in the study.**
(DOCX)

**S2 Table. Inbreeding data, the number of complete generations, and the number of incomplete (1 parent) generations in the animals under study.**
(DOCX)

## Acknowledgments

The authors would like to thank all the colleagues of the Osservatorio Veterinario Italiano Cardiopatie, and Doctor Luisa Ginoulhiac (Maple Tree breeding) for contributing to the collection of data presented in this study.

## Author Contributions

**Conceptualization:** Mara Bagardi, Michele Polli, Paola Giuseppina Brambilla.

**Data curation:** Mara Bagardi, Giulietta Minozzi, Giulia Drago, Elisa Lorenzi, Francesca Zanchi.

**Formal analysis:** Mara Bagardi, Sara Ghilardi, Giulietta Minozzi, Eleonora Fusi.

**Funding acquisition:** Mara Bagardi, Paolo Luigi Ferrari, Paola Giuseppina Brambilla.

**Investigation:** Mara Bagardi, Sara Ghilardi, Giulietta Minozzi, Eleonora Fusi, Paolo Luigi Ferrari, Michele Polli, Elisa Lorenzi, Francesca Zanchi.

**Methodology:** Mara Bagardi.

**Project administration:** Paolo Luigi Ferrari, Paola Giuseppina Brambilla.

**Resources:** Michele Polli, Paola Giuseppina Brambilla.

**Supervision:** Giulietta Minozzi, Eleonora Fusi, Chiara Locatelli, Paolo Luigi Ferrari, Paola Giuseppina Brambilla.

**Validation:** Mara Bagardi, Sara Ghilardi, Paola Giuseppina Brambilla.

**Visualization:** Francesca Zanchi.

**Writing – original draft:** Mara Bagardi, Elisa Lorenzi.

**Writing – review & editing:** Mara Bagardi, Sara Ghilardi, Giulietta Minozzi, Eleonora Fusi, Chiara Locatelli, Paolo Luigi Ferrari, Giulia Drago, Michele Polli, Elisa Lorenzi, Francesca Zanchi, Paola Giuseppina Brambilla.

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
