## [Decision Letter · Decision Letter 0]

22 Feb 2024

PONE-D-23-41720Study of correlations between serum taurine, thyroid hormones and echocardiographic parameters of systolic function in clinically healthy Golden retrievers fed with commercial dietPLOS ONE

Dear Dr. Minozzi,

Thank you for submitting your manuscript to PLOS ONE. After careful consideration, we feel that it has merit but does not fully meet PLOS ONE’s publication criteria as it currently stands. Therefore, we invite you to submit a revised version of the manuscript that addresses the points raised during the review process.

**ACADEMIC EDITOR: **The authors should provide point by point response to each question as all questions are relevant. Moreover, they should add number of protocol authorized from the national Ethical Committee.

We look forward to receiving your revised manuscript.

Kind regards,

Vincenzo Lionetti, M.D., PhD

Academic Editor

PLOS ONE

Journal Requirements:

https://www.researchgate.net/publication/341414940_Development_of_plasma_and_whole_blood_taurine_reference_ranges_and_identification_of_dietary_features_associated_with_taurine_deficiency_and_dilated_cardiomyopathy_in_golden_retrievers_A_prospective_o

https://journals.plos.org/plosone/article?id=10.1371%2Fjournal.pone.0233206

In your revision ensure you cite all your sources (including your own works), and quote or rephrase any duplicated text outside the methods section. Further consideration is dependent on these concerns being addressed.

3. Please be informed that funding information should not appear in the Acknowledgments section or other areas of your manuscript. We will only publish funding information present in the Funding Statement section of the online submission form. Please remove any funding-related text from the manuscript.

4. We notice that your supplementary table are included in the manuscript file. Please remove them and upload them with the file type 'Supporting Information'. Please ensure that each Supporting Information file has a legend listed in the manuscript after the references list.

Reviewers' comments:

Reviewer's Responses to Questions

**Comments to the Author**

1. Is the manuscript technically sound, and do the data support the conclusions?

Reviewer #1: Partly

Reviewer #2: Partly

2. Has the statistical analysis been performed appropriately and rigorously? 

Reviewer #1: Yes

Reviewer #2: N/A

3. Have the authors made all data underlying the findings in their manuscript fully available?

Reviewer #1: Yes

Reviewer #2: No

4. Is the manuscript presented in an intelligible fashion and written in standard English?

Reviewer #1: Yes

Reviewer #2: Yes

5. Review Comments to the Author

Reviewer #1: Summary of work in manuscript

This manuscript presents results of a cross-sectional study examining serum taurine levels with cardiac function in Golden Retriever dogs. The goal was to identify breed-specific reference ranges and a clinical cut-off for taurine levels that could be used to indicate likely adverse cardiac function in this breed. Other parameters such a CBC, blood chemistry, thyroid hormone and cTnI were also examined. This study confirms a relationship between hypotaurinemia and impaired systolic function as well as provides important novel information that is useful for veterinary clinicians in assessing causes or potential for dilated cardiomyopathy in Golden Retrievers. Overall, the manuscript is well written. While the study design and sample size are good, with good data analyses and presentation of echocardiography, the authors failed to present all the other blood parameters that were mentioned, some of which were discussed and some that were completely ignored. Moreover, some of the methods used were not included in the Methods section, including most critically the taurine analysis methods and assay performance. The following are specific items for the authors to consider further:

Specific comments

1. Line 27 and elsewhere in manuscript: The word ‘whit’ is incorrect. Replace with ‘with’.

2. Line 34: This first number of 398.2 is not the range (the range is in parentheses). Please indicate whether this is the median or the mean value.

3. Lines 135-136: Please provide a brief description, indicate the standard clinical analytical machine or cite a reference for methods for each of these blood parameters (CBC, biochemical analysis, T4 and TSH). Of these parameters, the most critical method that absolutely requires more detail is the method to measure taurine concentration. Since this is the focal point of this manuscript, a longer, more detailed analysis and provision of assay performance parameters similar to that given for the cTnI should be included here in the methods.

4. Results section: There is no mention, nor any tables or figure containing the data from the CBC, blood chemistry and cTnI results in this Results section. This data should at least be included in the Supplemental information. Then, the authors need to provide some statements regarding these results must be mentioned since they were part of the study. They are also important reassurances, even if no differences are present, that the dogs were otherwise healthy.

5. Line 292-293: The hypothesis that intestinal dysbiosis is altering taurine levels through altered absorption is either flawed or requires more explanation. The majority of the intestinal microbial community in dogs is located in the large intestine, a point at which there is poor amino acid and taurine absorption. Instead, the microbiome could be using taurine for anaerobic or other metabolic processes and consuming the taurine. This still would only affect fecal taurine levels, not so much the levels that the golden retrievers are able to absorb. Please rethink this hypothesis and come up with an alternative explanation or provide a reference that supports your hypothesis and explain it a bit better here.

Reviewer #2: The study investigates the relationship between serum taurine levels and left ventricular systolic function in Golden Retrievers, particularly focusing on potential connections to dilated cardiomyopathy (DCM). The research reveals a significant correlation between lower serum taurine concentrations and impaired systolic function. The findings suggest that monitoring taurine levels could be crucial in identifying dogs at risk of DCM. The study also explores dietary and thyroid factors but acknowledges limitations, emphasizing the need for larger and more diverse samples with follow-up data to strengthen the conclusions. The study is well structured and well written nevertheless clarifying statistical methods and addressing potential biases would enhance the study's reliability.

In particular, I have these main revisions to highlight:

- Though I understand how difficult it is to include this number of cases I believe that the power of the sample is not sufficient for a Pearson correlation study. In particular a few more cases would be needed to draw these conclusions with statistical significance. In the case that has been done I would ask for the power of the sample. In case this is not adequate for the study I would propose to extend the study by including the necessary cases or in case this is not possible I would propose to highlight this strong limitation leaving the study more descriptive and less statistical.

- I suggest including graphs of pearson correlations and ROC curves. This allows for better interpretation of the data described in both the results and discussions.

Minor revisions:

- Line 130: “with” instead of “whit”

- Line 133: Remove one of the two "."

- Line 136: I think it is "serum" taurine instead of "plasma" taurine.

- Line 208: "Systolic disfunction" should be "Systolic dysfunction"

- Line 132: "methionine or l-carnitine" should be "methionine or L-carnitine"

- Line 195: "normal ranges for the weight" should be "normal ranges for weight"

6. PLOS authors have the option to publish the peer review history of their article (what does this mean?). If published, this will include your full peer review and any attached files.

Reviewer #1: No

Reviewer #2: No

---

## [Author Response · Author response to Decision Letter 0]

28 Mar 2024

PONE-D-23-41720

Study of correlations between serum taurine, thyroid hormones and echocardiographic parameters of systolic function in clinically healthy Golden retrievers fed with commercial diet

The authors thank the Editor and the Reviewers for their thorough review of our study. 

We have carefully considered all Reviewers’ comments and have tried to address them whenever we felt this was appropriate. Furthermore, we have rephrased any duplicated text outside the methods section as suggested by the Editor thanks to a plagiarism check. We feel that the quality of our manuscript has improved following the Reviewers’ comments and suggestions.

Best regards

Review Comments to the Author

Reviewer #1: 

Summary of work in manuscript

This manuscript presents results of a cross-sectional study examining serum taurine levels with cardiac function in Golden Retriever dogs. The goal was to identify breed-specific reference ranges and a clinical cut-off for taurine levels that could be used to indicate likely adverse cardiac function in this breed. Other parameters such a CBC, blood chemistry, thyroid hormone and cTnI were also examined. This study confirms a relationship between hypotaurinemia and impaired systolic function as well as provides important novel information that is useful for veterinary clinicians in assessing causes or potential for dilated cardiomyopathy in Golden Retrievers. Overall, the manuscript is well written. While the study design and sample size are good, with good data analyses and presentation of echocardiography, the authors failed to present all the other blood parameters that were mentioned, some of which were discussed and some that were completely ignored. Moreover, some of the methods used were not included in the Methods section, including most critically the taurine analysis methods and assay performance. The following are specific items for the authors to consider further:

The authors thank the reviewer for the careful review of the study and the positive comments provided. Information related to blood parameters and the type of analysis have been added in the Results and Methods sections respectively. 

Specific comments

1. Line 27 and elsewhere in manuscript: The word whit is incorrect. Replace with “with”.

Thank you very much, there were two mistakes. Thank you, we have replaced them. 

2. Line 34: This first number of 398.2 is not the range (the range is in parentheses). Please indicate whether this is the median or the mean value.

Thank you for the comment. Our statistician suggested expressing the range as 398.2, which is the span between the minimum (31.8) and the maximum (430) values observed. The value 398.2 is neither the mean nor the median. In descriptive statistics, the range is the size of the smallest interval which contains all the data and provides an indication of statistical dispersion. Since it only depends on two of the observations, it is most useful in representing the dispersion of small data sets. For this reason, we decided to not express the mean or the median value. 

3. Lines 135-136: Please provide a brief description, indicate the standard clinical analytical machine or cite a reference for methods for each of these blood parameters (CBC, biochemical analysis, T4 and TSH). Of these parameters, the most critical method that absolutely requires more detail is the method to measure taurine concentration. Since this is the focal point of this manuscript, a longer, more detailed analysis and provision of assay performance parameters similar to that given for the cTnI should be included here in the methods.

Thank you for your comment. We agree with the reviewer, that information was lacking, and we have added the description of the type of laboratory analysis and the data of other blood parameters. 

4. Results section: There is no mention, nor any tables or figure containing the data from the CBC, blood chemistry and cTnI results in this Results section. This data should at least be included in the Supplemental information. Then, the authors need to provide some statements regarding these results must be mentioned since they were part of the study. They are also important reassurances, even if no differences are present, that the dogs were otherwise healthy.

Thank you for this comment. We have added these data to the Results section, and we have created a table included as supporting information. In the discussion has been added a comment related to the blood work results. 

5. Line 292-293: The hypothesis that intestinal dysbiosis is altering taurine levels through altered absorption is either flawed or requires more explanation. The majority of the intestinal microbial community in dogs is located in the large intestine, a point at which there is poor amino acid and taurine absorption. Instead, the microbiome could be using taurine for anaerobic or other metabolic processes and consuming the taurine. This still would only affect fecal taurine levels, not so much the levels that the golden retrievers are able to absorb. Please rethink this hypothesis and come up with an alternative explanation or provide a reference that supports your hypothesis and explain it a bit better here.

Thank you for your comment. To our knowledge studies on the correlation of serum taurine with intestinal dysbiosis in dogs are lacking. We agree with your suggestion. Loss through faeces due to a lack of deconjugation and reabsorption or microbial consumption are some factors that could partially influence taurine concentration. However, in human medicine studies between alterations in the gut microbiota and serum taurine concentration are present and suggest further roles for this amino acid.

Reviewer #2: 

The study investigates the relationship between serum taurine levels and left ventricular systolic function in Golden Retrievers, particularly focusing on potential connections to dilated cardiomyopathy (DCM). The research reveals a significant correlation between lower serum taurine concentrations and impaired systolic function. The findings suggest that monitoring taurine levels could be crucial in identifying dogs at risk of DCM. The study also explores dietary and thyroid factors but acknowledges limitations, emphasizing the need for larger and more diverse samples with follow-up data to strengthen the conclusions. The study is well structured and well written nevertheless clarifying statistical methods and addressing potential biases would enhance the study’s reliability.

The authors thank Reviewer 2 for the careful review of the study and the comments provided. The potential biases underlined by the Reviewer have been evaluated and checked by our statistician (Professor Annamaria Zanaboni). Below are the comments provided. 

In particular, I have these main revisions to highlight:

- Though I understand how difficult it is to include this number of cases I believe that the power of the sample is not sufficient for a Pearson correlation study. In particular a few more cases would be needed to draw these conclusions with statistical significance. In the case that has been done I would ask for the power of the sample. In case this is not adequate for the study I would propose to extend the study by including the necessary cases or in case this is not possible I would propose to highlight this strong limitation leaving the study more descriptive and less statistical.

Thank you for your comments and suggestions. The power of our sample is 80% in detecting an effect size greater than 0.35, and we agree that a few additional cases would be necessary to detect at least a moderate effect size (that is a correlation absolute value greater than 0.30). Since it isn’t possible to add new cases to the study, we changed the text accordingly.

- I suggest including graphs of pearson correlations and ROC curves. This allows for better interpretation of the data described in both the results and discussions.

Thank you for your suggestions. We have added in the Results section, as figures, the ROC curve of the taurine cut-off for the identification of the impaired systolic function and the scatter plots of variables whose correlation absolute value with serum taurine was higher than 0.35. We agree with the Reviewer that they allow better interpretation of the data. 

Minor revisions:

- Line 130: “with” instead of “whit”;

Thank you, there was a mistake. We have corrected it. 

- Line 133: Remove one of the two “.”;

Thank you very much. We have deleted one of them. 

- Line 136: I think it is "serum" taurine instead of "plasma" taurine.

Thank you for the comment, yes it was a mistake. We have corrected it.

- Line 208: "Systolic disfunction" should be "Systolic dysfunction";

Thank you, we have corrected the mistake.

- Line 132: "methionine or l-carnitine" should be "methionine or L-carnitine”;

Thank you, we have corrected the mistake. 

- Line 195: "normal ranges for the weight" should be "normal ranges for weight";

Thank you for the suggestion. We have deleted “the”. 

---

## [Decision Letter · Decision Letter 1]

10 Apr 2024

Study of correlations between serum taurine, thyroid hormones and echocardiographic parameters of systolic function in clinically healthy Golden retrievers fed with commercial diet

PONE-D-23-41720R1

Dear Dr. Minozzi,

We’re pleased to inform you that your manuscript has been judged scientifically suitable for publication and will be formally accepted for publication once it meets all outstanding technical requirements.

Kind regards,

Vincenzo Lionetti, M.D., PhD

Academic Editor

PLOS ONE

Additional Editor Comments (optional):

Reviewers' comments:

Reviewer's Responses to Questions

**Comments to the Author**

1. If the authors have adequately addressed your comments raised in a previous round of review and you feel that this manuscript is now acceptable for publication, you may indicate that here to bypass the “Comments to the Author” section, enter your conflict of interest statement in the “Confidential to Editor” section, and submit your "Accept" recommendation.

Reviewer #1: All comments have been addressed

2. Is the manuscript technically sound, and do the data support the conclusions?

Reviewer #1: Yes

3. Has the statistical analysis been performed appropriately and rigorously? 

Reviewer #1: Yes

4. Have the authors made all data underlying the findings in their manuscript fully available?

Reviewer #1: (No Response)

5. Is the manuscript presented in an intelligible fashion and written in standard English?

Reviewer #1: Yes

6. Review Comments to the Author

Reviewer #1: The revisions performed addressed all concerns this reviewer had. The authors should do a check of grammar for the new additions, but other than that, I have nothing to add.

7. PLOS authors have the option to publish the peer review history of their article (what does this mean?). If published, this will include your full peer review and any attached files.

Reviewer #1: No

---

## [Editor Report · Acceptance letter]

2 May 2024

PONE-D-23-41720R1 

PLOS ONE

Dear Dr. Minozzi, 

I'm pleased to inform you that your manuscript has been deemed suitable for publication in PLOS ONE. Congratulations! Your manuscript is now being handed over to our production team.

Kind regards, 

on behalf of

Prof. Vincenzo Lionetti 

Academic Editor

PLOS ONE